# BLOCK COORDINATE DESCENT FOR NEURAL NETWORKS PROVABLY FINDS GLOBAL MINIMA

## ABSTRACT

In this paper, we consider a block coordinate descent (BCD) algorithm for training deep neural networks and provide a new global convergence guarantee under strictly monotonically increasing activation functions. While existing works demonstrate convergence to stationary points for BCD in neural networks, our contribution is the first to prove convergence to global minima, ensuring arbitrarily small loss. We show that the loss with respect to the output layer decreases exponentially while the loss with respect to the hidden layers remains well-controlled. Additionally, we derive generalization bounds using the Rademacher complexity framework, demonstrating that BCD not only achieves strong optimization guarantees but also provides favorable generalization performance. Moreover, we propose a modified BCD algorithm with skip connections and non-negative projection, extending our convergence guarantees to ReLU activation, which are not strictly monotonic. Empirical experiments confirm our theoretical findings, showing that the BCD algorithm achieves a small loss for strictly monotonic and ReLU activations.

## 1 INTRODUCTION

Deep learning has led to significant advances across various domains, such as computer vision, natural language processing, and reinforcement learning, achieving unprecedented performance in numerous tasks. However, understanding the training dynamics and optimization behavior of deep neural networks remains an ongoing challenge due to the highly non-convex nature of their loss functions (Li et al., 2018). Proving convergence to global minima of gradient descent via backpropagation, particularly for deep networks with multiple layers, remains an open problem in the field. While the neural tangent kernel (NTK) regime (Jacot et al., 2018) addresses this problem by reducing the non-convex loss to the convex one in RKHS, it fails to fully explain the empirical success of deep learning because it often outperforms kernel methods, even if we employ NTK as the kernel.

Contrary to the backpropagation-based training, the block coordinate descent (BCD), which originated from the mathematical optimization field (see Tseng (2001), for example), is an optimization framework where we divide a variable into several blocks and optimize them alternately. BCD offers computational advantages by updating subsets of parameters iteratively, allowing for tractable optimization of complex systems. The objective function appearing in the neural network training is also highly non-convex, and to overcome this issue, BCD-based neural network optimization methods have been proposed (Carreira-Perpinan & Wang, 2014; Askari et al., 2018; Lau et al., 2018; Zhang & Brand, 2017; Patel et al., 2020; Zeng et al., 2019; Nakamura et al., 2021; Qiao et al., 2021; Zhang et al., 2022; Xu et al., 2024). When we apply BCD to neural network training, the most natural way is that we regard the weights of each layer as a block, and existing works adopt this way. By the formulation of BCD, the loss function of the neural network can be divided into several components, one of which coincides with a loss with respect to a layer. Compared to the original loss, these divided ones have more accessible landscapes to optimize.

Based on such an advantage of BCD for neural networks, its theoretical perspective, mainly about its convergence guarantee, has been explored in recent years. However, existing theoretical works on BCD for neural networks (Zhang & Brand, 2017; Zeng et al., 2019; Zhang et al., 2022; Xu et al., 2024) have only focused on the convergence to stationary points, points with zero gradients. Convergence to stationary points does not imply convergence to global minima, especially when

the objective function is highly non-convex, such as the loss that appears in the training of neural networks (Li et al., 2018; Safran & Shamir, 2018).

How neural network training finds global minima has been one of the most significant topics in deep learning theory literature. However, existing guarantees on BCD remain in convergence to the stationary points. To bridge this gap, we aim to provide the convergence guarantee to the global minima of BCD for neural networks. To this end, we consider multi-layer neural networks and employ a BCD-type algorithm, updating the parameters using vanilla gradient descent. Our contribution can be summarized as follows:

- We prove the global convergence of a block coordinate descent (BCD) algorithm, where we train deep neural network models with strictly monotonically increasing activation. We ensure that the parameters attain arbitrarily small loss by proving that (i) the loss with respect to the output layer will decrease exponentially to zero and (ii) the loss with respect to the hidden layers remains small in every iteration. Through the analysis, we carefully evaluate the difference propagated from the output layer to the input layer. To the best of our knowledge, this is the first result that guarantees convergence to the global minima of neural networks with any number of layers beyond the NTK regime.

- We derive a generalization error bound of deep neural networks trained by BCD under settings with i.i.d. data. In the convergence analysis, we show that the norm of weight matrices of each layer can be bounded by a constant. Combining this and the Rademacher complexity argument from Bartlett et al. (2017) gives a upper bound on generalization error. Compared to the existing works on gradient descent, BCD enables us to provide the generalization gap bound for multi-layer neural networks with an optimization guarantee.

- A notable challenge in applying our approach to commonly used activation functions like ReLU is their non-monotonic nature. Since ReLU is not strictly monotonically increasing, our initial convergence result does not directly apply. To address this issue, we propose a modified BCD algorithm incorporating skip connections (He et al., 2016) and non-negative projection updates. This modification ensures that convergence guarantees extend to ReLU networks, thereby broadening the applicability of our method to real-world architectures that predominantly use ReLU activations.

- We validate our theoretical findings through numerical experiments, showing that BCD for both strictly monotonic and ReLU activations achieves arbitrarily small loss values. These empirical results confirm the practical viability of our proposed methods, demonstrating their effectiveness in optimizing deep neural networks beyond theoretical guarantees.

## 1.1 OTHER RELATED WORKS

**Convergence guarantee of GD/SGD for neural networks**   In recent years, theoretical works on the convergence guarantee of (stochastic) gradient descent for neural networks have been intensively investigated. In the neural tangent kernel (NTK) regime (Jacot et al., 2018; Allen-Zhu et al., 2019b; Arora et al., 2019; Du et al., 2019; Zou et al., 2020), to name a few, the training dynamics of deep neural networks can be approximated by the gradient descent in RKHS. While we can ensure its global convergence by exploiting the convexity, the *feature learning ability* of neural networks, which is considered one of the critical ingredients of the practical success of deep learning, is not reflected since the training dynamics are reduced to the kernel method. For example, the parameters of networks trained by The NTK regime hardly move from their initial points as the number of parameters increases. On the other hand, our analysis does not fall into such a situation. Moreover, our analysis does not require any *overparameterization* on hidden layers to ensure global convergence.

The mean-field (MF) regime (Nitanda & Suzuki, 2017; Chizat & Bach, 2018; Mei et al., 2019; Tzen & Raginsky, 2020; Pham & Nguyen, 2021; Nguyen & Pham, 2023) is another promising approach of investigating neural network training. It regards the training of parameters as that of (probability) measure over the parameters, by which we can convert the non-convex optimization with respect to the parameters to the convex one where the distribution of parameters itself is a variable to be optimized. While several studies ensure its global convergence by employing this convexity without loss of feature learning ability, most of their models only focus on two or three-layer networks, our analysis admits the any number of hidden layers.

More recently, Banerjee et al. (2023) proposed restricted strong convexity (RSC) to analyze neural network training, which derives the global convergence guarantee by assuming that the gradient and output of neural networks correlate with each other during the training. However, Banerjee et al. (2023) still requires an analysis of this correlation assumption and does not fully explain the nature of global convergence in the training.

**Generalization error bound of multi-layer neural networks** Investigation of generalization error analysis for multi-layer neural networks has been explored in recent years (Neyshabur et al., 2015; Wei & Ma, 2019; Bartlett et al., 2017; Neyshabur et al., 2017; Golowich et al., 2018; Bartlett et al., 2019; Arora et al., 2018; Suzuki et al., 2020). These works give a generalization error by evaluating the complexity of neural networks from various perspective, such as the VC-dimension, the norm of parameters of networks, and so on. On the other hand, most of these results do not consider the optimization, but we also demonstrate the global convergence guarantee. Moreover, several works on generalization error analysis go beyond two-layer networks. However, most focus only on three-layer networks (Allen-Zhu & Li, 2019; Allen-Zhu et al., 2019a).

## 2 PRELIMINARIES

### 2.1 NOTATIONS

For an integer $n$, we define $[n] \coloneqq \{1, \ldots, n\}$. For $x \in \mathbb{R}^d$, $\|x\|$ denotes its Euclidean norm. We denote the $d$-dimensional identity matrix by $I_d$. For $A \in \mathbb{R}^{n \times m}$, $\|A\|_F \coloneqq \sqrt{\sum_{i,j} A_{ij}^2}$ denotes its Frobenius norm, and $\|A\|_{op} \coloneqq \max_{\|x\| \leq 1} \|Ax\|$ denotes its operator norm. For two symmetric matrices $A$ and $B$, we denote $A \prec B$ ($A \preceq B$) if and only if the matrix $B - A$ is positive (non-negative) definite. For $x = (x_1, \ldots, x_d)^\top \in \mathbb{R}^d$, $\mathrm{diag}(x) \in \mathbb{R}^{d \times d}$ denotes a diagonal matrix whose $j$-th diagonal component is $x_j$.

### 2.2 PROBLEM SETTINGS

Here, we introduce problem settings we consider in this paper. We observe $n$ training examples $\mathcal{D} = \{(x_i, y_i)\}_{i=1}^n$, where $x_i \in \mathbb{R}^{d_{in}}$ is a feature vector and $y_i \in \mathbb{R}^{d_{out}}$ is a label. Let $X = (x_1 \ldots x_n)^\top \in \mathbb{R}^{n \times d_{in}}$. Throughout the analysis, we consider high-dimensional settings $n \leq d_{in}$. Moreover, we make an assumption about the matrix $X$ as follows:

**Assumption 1** (Data matrix is full row rank). $\mathrm{rank}(X) = n$.

This assumption is required to show the global convergence. As we will see in the proof of the main result, we cannot ensure the existence of global minima without Assumption 1.

A multi-layer neural network is defined by

$$f_{NN}(x) \coloneqq W_L \sigma(W_{L-1} \sigma(\ldots W_2 \sigma(W_1 x)) \ldots),$$

where $\sigma$ is element-wise activation and $W_1 \in \mathbb{R}^{r \times d_{in}}$, $W_j \in \mathbb{R}^{r \times r}$ for $j \in \{2, \ldots, L-1\}$, and $W_L \in \mathbb{R}^{d_{out} \times r}$. We consider that all the hidden layers have the same width $r$.

Then, we make the following assumption on the activation function.

**Assumption 2** (Activation). $\sigma : \mathbb{R} \to \mathbb{R}$ *is monotonically increasing and satisfies* $\sigma(0) = 0$. *Especially, there exists a constant* $0 < \alpha < 2$ *such that* $\inf_{x \in \mathbb{R}} \sigma'(x) \geq \alpha$ *holds*[1]. *Moreover,* $\sigma$ *is* $\ell$*-Lipschitz, i.e., for any* $u_1, u_2 \in \mathbb{R}$, $|\sigma(u_1) - \sigma(u_2)| \leq \ell|u_1 - u_2|$ *holds.*

A typical example of activation function satisfying Assumption 2 is LeakyReLU activation $x \mapsto \max\{x, ax\}$ ($a < 1$): which satisfies Assumption 2 with $\alpha = a$ and $\ell = 1$. We note that other activation, such as ReLU $x \mapsto \max\{x, 0\}$, does not satisfy Assumption 2. We also provide the global convergence algorithm when we use the ReLU activation in Section 5.

Under this formulation of neural networks, we formalize the regression problem

$$\min_{\mathbf{W}} \sum_{i=1}^n (f_{NN}(x_i) - y_i)^2, \tag{1}$$

---

[1]If $\sigma$ is not differentiable, we assume that $\sigma(x_1) - \sigma(x_2) \geq \alpha(x_1 - x_2)$ for any $x_1, x_2 \in \mathbb{R}$.

where $\mathbf{W} = (W_1, \ldots, W_L)$. One of the most straightforward approaches to solve (1) is (stochastic) gradient method, in which the parameters are updated using the loss gradient. Conversely, we employ a layer-wise optimization method called *block coordinate descent*, as we introduce in the following section.

## 3 BLOCK COORDINATE DESCENT

In this section, after we introduce the basic notion of the *block coordinate descent* (BCD), we provide the algorithm we consider in this paper. BCD, which originated from the mathematical optimization field (see Tseng (2001), for example), is an optimization framework where we divide a variable into several blocks and optimize them alternately.

In BCD, instead of directly utilizing the loss (1), we introduce auxiliary parameters $V_{1,i} \ldots V_{L,i}$. $V_{j,i}$ aims to approximate the output of $j$-th layer for the $i$-th sample $x_i$. By construction, we have $V_{j,i} \in \mathbb{R}^r$ for $j = 1, \ldots, L-1$. By using these auxiliary parameters, we reformulate (1) as follows:

$$\min_{\mathbf{W},\mathbf{V}} F(\mathbf{W}, \mathbf{V}) := \sum_{i=1}^n \left[ \|W_L V_{L-1,i} - y_i\|^2 + \gamma \sum_{j=1}^{L-1} \|\sigma(W_j V_{j-1,i}) - V_{j,i}\|^2 \right], \quad (2)$$

where $\gamma > 0$ is a hyperparameter and we denote $V_{0,i} := x_i$, $\mathbf{W} = (W_1, \ldots, W_L)$, and $\mathbf{V} = (V_{1,1}, \ldots, V_{L-1,n})$. In the reformulated problem (2), the second term represents the loss at the $j$-th layer, indicating how $V_{j,i}$ approximates the output of the layer given the input $x_i$. The first term represents the loss at the output layer, showing how close the outputs of the network with the approximated (hidden layer) output $V_{j,i}$ are to the training labels $y_1, \ldots, y_n$. By the construction, if $(\mathbf{W}^*, \mathbf{V}^*)$ satisfies $F = 0$ in (2), $\mathbf{W}^*$ is the optimal solution of (1).

One of the benefits of the reformulation (2) is that we can treat the objective function with respect to the weights of each layer $(W_1, \ldots, W_L)$ separately. Such a simplification results not only in a faster implementation (e.g., parallelization) but also a favorable loss landscape, including theoretical tractability. While various methods for optimizing (2) have been explored, we consider a relatively simple one, updating weights $W_j$ and auxiliary variables $V_{j,i}$ sequentially from the output layer. Specifically, we update the variables in order $W_L \to V_{L-1,i} \to W_{L-1} \to \ldots V_{1,i} \to W_1$ by using the objective function (2). We summarize the algorithm considered in this paper in Algorithm 1. From now on, we explain its detailed procedure.

---

**Algorithm 1:** Block coordinate descent

**input** : $(W_1)_{ab} \overset{i.i.d.}{\sim} \mathcal{N}(0, 1/d_{in})$, $(W_j)_{ab} \overset{i.i.d.}{\sim} \mathcal{N}(0, 1/r)$ for all $j = 2, \ldots, L$, $V_{0,i} = x_i$.

1 $K$: max outer iteration, $K_V, K_W$: max inner iteration, $\eta_V, \eta_W^{(1)}, \eta_W^{(2)}$: step size;
2 $W_j \leftarrow$ output of Algorithm 2 with inputs $s_1, s_2$, and $W_j$ for $j = 2, \ldots, L$;
3 $V_{j,i} \leftarrow \sigma(W_j V_{j-1,i})$ for all $j = 1, \ldots, L-1$ and $i = 1, \ldots, n$.;
4 **for** $k \leftarrow 1$ **to** $K$ **do**
5 $\quad$ $W_L \leftarrow W_L - \eta_W^{(1)} \nabla_{W_L} \sum_{i=1}^n \|W_L V_{L-1,i} - y_i\|^2$;
6 $\quad$ **for** $i \leftarrow 1$ **to** $n$ **do**
7 $\quad\quad$ $V_{L-1,i} \leftarrow V_{L-1,i} - \eta_V \nabla_{V_{L-1,i}} \|W_L V_{L-1,i} - y_i\|^2$;
8 $\quad$ **for** $j \leftarrow L-1$ **to** $2$ **do**
9 $\quad\quad$ $W_j \leftarrow W_j - \gamma \eta_W^{(1)} \sum_{i=1}^n \nabla_{W_j} \|\sigma(W_j V_{j,i}) - V_{j+1,i}\|^2$;
10 $\quad\quad$ **for** $i \leftarrow 1$ **to** $n$ **do**
11 $\quad\quad\quad$ **for** $k_{inner} \leftarrow 1$ *to* $K_V$ **do**
12 $\quad\quad\quad\quad$ $V_{j-1,i} \leftarrow V_{j-1,i} - \gamma \eta_V \nabla_{V_{j-1,i}} \|\sigma(W_j V_{j-1,i}) - V_{j,i}\|^2$ ;
13 $\quad$ **for** $k_{inner} \leftarrow 1$ **to** $K_W$ **do**
14 $\quad\quad$ $W_1 \leftarrow W_1 - \gamma \eta_W^{(2)} \sum_{i=1}^n \nabla_{W_1} \|\sigma(W_1 V_{0,i}) - V_{1,i}\|^2$ ;

---

**Initialization** We consider Gaussian initialization for $W_j$; that is, each element of $W_1$ is sampled from $\mathcal{N}(0, d_{in}^{-1})$, and each element of $W_j$ $(j = 2, \ldots, L)$ is sampled from $\mathcal{N}(0, r^{-1})$. After that, we apply *singular value bounding* (SVB) (Jia et al., 2017) to $W_j$ $(j = 2, \ldots, L)$. In SVB, we

---

**Algorithm 2:** Singular Value Bounding

**input** : $W_j$: matrix, $(s_1, s_2)$: lower and upper bounds on the singular values
1 $(U, \Sigma, V) \leftarrow$: Singular value decomposition of $W_j = U\Sigma V$;
2 **for** $s \leftarrow$ *diagonal components of* $\Sigma$ **do**
3 $\quad | \quad s \leftarrow \max\{s_1, \min\{s_2, s\}\}$;
**output:** $U\Sigma V$

---

conduct the singular value decomposition of $W_j$ as $W_j = U\Sigma V^\top$, where $U$ and $V$ are orthogonal matrices, and $\Sigma$ is a non-negative diagonal matrix. Since $W_j$ is full-rank with probability 1 over the initialization, we also have $\Sigma \in \mathbb{R}^{r \times r}$ with probability 1. After SVB, we adjust each diagonal component of $\Sigma$ to be within the interval $[s_1, s_2]$. Then, we utilize $W_j = U\Sigma'V^\top$ as the initial parameter of $W_j$, where $\Sigma'$ be the matrix obtained by the adjustment. We summarize this procedure in Algorithm 2.

In Jia et al. (2017), SVB is conducted at every epoch to enhance the stability of the training and prediction performance of stochastic gradient descent. The upper and lower bounding of the singular value prevents the amplifying or vanishing of a gradient in the backpropagation. Applying SVB also has several advantages in BCD, not only for practical reasons but also from a theoretical perspective. First, the regularity of $W_j$ results in a preferable condition number of the objective function $\|\sigma(W_j V_{j-1,i}) - V_{j,i}\|^2$ in $F$, the loss at the $j$-th layer. Moreover, the upper bound on the singular value prevents $V_j$ from becoming extremely large at the initialization.

**Remark 3.1.** *While Jia et al. (2017) applies SVB at every epoch, we use it only at the initialization. By setting the step size not too large, we can ensure that all the singular values of $W_j$ remain in a bounded interval, as we show in the proof, with which we enjoy the same benefit throughout the training.*

After initializing $W_j$, we initialize $V_j$ in an exact manner, i.e., $V_{j,i} = \sigma(W_j V_{j-1,i})$ for all $j = 1, \ldots, L-1$ and $i = 1, \ldots, n$. While we can employ any initialization scheme for $V_j$, the exact manner results in $\|\sigma(W_j V_{j-1,i}) - V_{j,i}\|^2 = 0$ at the initialization, leading to faster convergence.

**Update of** $V$ For optimizing $W$ and $V$, we utilize vanilla gradient descent. We employ a common step size $\eta_V$ for each $V_{j,i}$ and perform multiple updates using the loss $\|\sigma(W_j V_{j-1,i}) - V_{j,i}\|^2$ (line 6, 12), given by

$$V_{L-1,i} \leftarrow V_{L-1,i} - \eta_V \nabla_{V_{L-1,i}} \|W_L V_{L-1,i} - y_i\|^2 \tag{3}$$

and

$$V_{j-1,i} \leftarrow V_{j-1,i} - \gamma\eta_V \nabla_{V_{j-1,i}} \|\sigma(W_j V_{j-1,i}) - V_{j,i}\|^2.$$

The first update (3) can be interpreted as solving the linear equation $W_L V_{L-1,i} = y_i$, which has a solution if the matrix $W_L$ is full row rank. We assume that the activation satisfies Assumption 2. In this case, since the mapping $\sigma : \mathbb{R} \to \mathbb{R}$ is a bijection, there exists an inverse map $\sigma^{-1}$, and training $V_j$ can be viewed as equivalent to solving the linear equation $W_{j+1} V_{j,i} = \sigma^{-1}(V_{j+1,i})$. Therefore, it is expected that $V_{j,i}$ converges to the solution via gradient descent with a suitable choice of $\eta_V$ as long as the matrix $W_j \in \mathbb{R}^{r \times r}$ is regular.

**Update of** $W$ For the update of $W_j$ $(j = 1, \ldots, L)$, we use the loss function at $j$-th layer, that is, $\sum_{i=1}^n \|W_L V_{L-1,i} - y_i\|^2$ for $W_L$ and $\sum_{i=1}^n \|\sigma(W_j V_{j-1,i}) - V_{j,i}\|^2$ for $W_j$ $(j = 1, \ldots, L-1)$.

For $W_2, \ldots, W_L$, we use a common step size $\eta_W^{(1)}$ and conduct the gradient descent update:

$$W_L \leftarrow W_L - \eta_W^{(1)} \nabla_{W_L} \sum_{i=1}^n \|W_L V_{L-1,i} - y_i\|^2,$$

and

$$W_j \leftarrow W_j - \gamma\eta_W^{(1)} \sum_{i=1}^n \nabla_{W_j} \|\sigma(W_j V_{j,i}) - V_{j+1,i}\|^2$$

for each iteration (line 5, 9). For $W_1$, we employ a different step size $\eta_W^{(2)}$ and apply

$$W_1 \leftarrow W_1 - \gamma\eta_W^{(2)} \sum_{i=1}^n \nabla_{W_1} \|\sigma(W_1 V_{0,i}) - V_{1,i}\|^2,$$

multiple ($K_W$) times for each iteration (line 14). These update manners are required to attain the global convergence. With respect to the loss of the second to $L$-th layer, we update both $W_j$ and $V_{j-1,i}$. In particular, by applying multiple updates to $V_{j-1,i}$, we can ensure linear convergence of the loss $\sum_{i=1}^{n}\|\sigma(W_j V_{j-1,i}) - V_{j,i}\|^2$ for each iteration while the singular values of matrix $W_j$ are upper and lower bounded. On the other hand, the existence of $W^*$ satisfying $\sum_{i=1}^{n}\|\sigma(W^* V_{j-1,i}) - V_{j,i}\|^2 = 0$ is not ensured, particularly in the case where $n > r$. Hence, it is not necessary to update $W_j$ for multiple times. Furthermore, as the number of iterations increases, it becomes less likely to maintain the regularity of the matrix $W_j$. This is why we only apply gradient descent once to $W_j$ ($j = 2, \ldots, L$). On the other hand, in the first layer, the input $V_{0,i} = x_i$ is fixed, and we need to demonstrate linear convergence of the loss $\sum_{i=1}^{n}\|\sigma(W_1 V_{0,i}) - V_{1,i}\|^2$ through the update of $W_1$. In the overparameterized setting $d_{in} \geq n$, if the data matrix satisfies $\mathrm{rank}(U) = n$, we can ensure the existence of a global minima $W^*$ satisfying $\sum_{i=1}^{n}\|\sigma(W^* V_{0,i}) - V_{1,i}\|^2 = 0$, and hence linear convergence under a suitable choice of $\eta_W^{(2)}$.

**Remark 3.2.** *Concerning the recent progress of the block coordinate descent algorithms applied to deep learning, as represented by (Jia et al., 2017; Zhang & Brand, 2017; Lau et al., 2018; Patel et al., 2020), among others, we employ a relatively simple approach using vanilla gradient descent without any regularization, focusing on devising the loss function and the order in which the parameters are updated. While our convergence proof is based on this specific setup, our analysis can be extended to encompass more complex scenarios. Our algorithm is adaptable to different settings, including potential applications to other loss functions and problems, such as classification problems, and the inclusion of regularization terms. We discuss possible extensions in Appendix A.*

## 4 GLOBAL CONVERGENCE OF BLOCK COORDINATE DESCENT

In this section, we show that BCD for neural networks with an activation satisfying Assumption 2 finds global minima, in other words, the objective value $F$ converges to an arbitrarily small value. In this section, we consider the case with single output ($d_{out} = 1$). We discuss its extension to the multi-output case in Appendix B. Moreover, for the single output case, we provide a bound on the generalization error under the i.i.d. setting by utilizing the Rademacher complexity argument.

### 4.1 GLOBAL CONVERGENCE WITH MONOTONICALLY INCREASING ACTIVATION

Here, we consider the case of single outputs $d_{out} = 1$. In this case, the objective function is described by

$$\min_{\mathbf{W},\mathbf{V}} F(\mathbf{W},\mathbf{V}) \coloneqq \sum_{i=1}^{n}\left[(W_L V_{L-1,i} - y_i)^2 + \gamma \sum_{j=1}^{L-1}\|\sigma(W_j V_{j-1,i}) - V_{j,i}\|^2\right]. \tag{4}$$

We now state the first main result, the global convergence of BCD with activation satisfying Assumption 2.

**Theorem 4.1** (BCD finds global minima of neural networks). *We assume that activation $\sigma$ satisfies Assumption 2 and there exists a constant $C_V > 0$ such that $\lambda_{\max}(V_j V_j^\top) \leq C_V$ for $j = 1, \ldots, L-1$ during training. We denote $s \coloneqq \sigma_{\min}(X) > 0$. Let $R_i = |W_L V_{L-1,i} - y_i|$ at the initial value of the objective function with respect to the output layer, and define $R \coloneqq \sum_{i=1}^{n} R_i^2$, $R_{\max} \coloneqq \max_i R_i$, and $C_K \coloneqq \left(\frac{2}{\alpha}\right)^L \left(4R_{\max}\eta_V + \frac{2}{2-\alpha}\sqrt{\epsilon}\right)$.*

*Then, under $(s_1, s_2) = (\frac{3}{4}, \frac{5}{4})$, $\eta_V \leq \frac{1}{8\alpha\ell^2}$, $\eta_W^{(1)} \leq \frac{\eta_V^{-1}}{8\sqrt{r}C_V K}\left(\frac{\alpha}{2}\right)^L$, $\eta_W^{(2)} \leq \frac{1}{2\ell^2 \cdot \max_i \|x_i\|}$, and*

$$K = \left\lceil \frac{2}{\eta_V}\log\left(\frac{3R}{\epsilon}\right)\right\rceil, K_V = \left\lceil \frac{1}{\gamma\alpha\ell\eta_V}\log\left(\frac{3\gamma(L-2)rnC_K^2}{\epsilon}\right)\right\rceil, K_W = \left\lceil \frac{1}{4\gamma s\alpha^2\eta_W^{(2)}}\log\left(\frac{3rnC_K^2}{\epsilon}\right)\right\rceil,$$

*it holds $F(\mathbf{W},\mathbf{V}) \leq \epsilon$, where $\mathbf{W} = (W_1, \ldots, W_L)$ and $\mathbf{V} = (V_{1,1}, \ldots, V_{L-1,n})$ are the parameters obtained by the output of Algorithm 1.*

The proof can be see in Appendix C. Theorem 4.1 exhibits that BCD provably finds a global minimum under a suitable choice of hyperparameters. While the definitions of $K$, $K_V$ and $K_W$ are

somewhat complex, the total number of gradient computation to achieve $\epsilon$ error is bounded by $\tilde{O}(K(LK_V + K_W)) = \tilde{O}(\log^2(\frac{1}{\epsilon}))$.

The proof consists of two parts: (i) the loss with respect to the output layer is monotonically decreasing in the outer loop, and (ii) the loss with respect to the hidden layer remains sufficiently small at the end of each iteration. We provide more detail to Appendix C due to page limitations.

We should note that the claims presented in Theorem 4.1 lie outside the framework of the so-called NTK regime (Jacot et al., 2018), among others. Specifically, while the NTK regime assumes that the parameters of neural networks remain almost unchanged during training, our analysis allows for scenarios where the parameters undergo changes of $\Omega(1)$.

**Remark 4.2.** *The assumption in Theorem 4.1, $\lambda_{\max}(V_j V_j^\top) \leq C_V$, ensures that the auxiliary parameters $V_{j,i}$ are bounded during training. While we assume the existence of $C_V$ in Theorem 4.1, we can provide a quantitative bound on the $C_V$ as $C_V = \mathcal{O}((\gamma \eta_V \ell n K K_V)^2)$ (note that this bound may not be tight). We provide a detailed derivation of this bound in Appendix D.*

### 4.2 GENERALIZATION ERROR BOUND

The objective of this subsection is to show that BCD Algorithm 1 does not only have a strong convergence guarantee, but also attains favorable generalization performance. To this end, we need to make an assumption about the data distribution.

**Assumption 3.** *The training sample $\{(x_i, y_i)\}_{i=1}^n$ is independently sampled from a distribution $(x, y) \sim P$. Under the distribution $P$, it holds that $\|x\| \leq B_X$ and $|y| \leq B_Y$ almost surely.*

The first statement defines the data distribution, which is essential and standard requirement for describing the generalization error. The one requires that inputs and labels should be bounded with probability one, which is also standard.

We then provide the following result on the generalization error bound.

**Theorem 4.3** (Generalization error bound). *Let $\hat{f}_{NN}$ be the output of Algorithm 1 under the same condition as Theorem 4.1. Then, if Assumption 3 holds,*

$$\underset{(x,y) \sim P}{\mathbb{E}}\left[\left(\hat{f}_{NN}(x) - y\right)^2\right] \leq \frac{1}{n} \sum_{i=1}^n \left(\hat{f}_{NN}(x_i) - y_i\right)^2$$

$$+ \tilde{O}\left(\frac{\|X\|}{n}(B_Y + 2^L \ell^{L-1} B_X) d_{in}^{\frac{1}{2}} L^{\frac{3}{2}} (2r)^{\frac{L}{2}} \log r + (B_Y + 2^L \ell^{L-1} B_X)^2 \sqrt{\frac{\log(1/\delta)}{n}}\right).$$

*with probability at least $1 - \delta$ over the training sample $\{(x_i, y_i)\}_{i=1}^n$.*

The proof can be seen in Appendix E. Notably, Theorem 4.3 provides a bound on the generalization error for multi-layer neural networks with optimization guarantees, beyond the NTK regime. To obtain Theorem 4.3, we utilize a result from Bartlett et al. (2017), which evaluates the generalization gap using the spectral norms of the weight matrix of each layer. As mentioned in the previous section, we can show that the spectral norm (equal to the maximum singular value) of $W_j$ is upper bounded. Combining this with the result from Bartlett et al. (2017), we can derive the generalization gap of BCD (see Appendix E for details).

## 5 RELU ACTIVATION

In this section, we propose a BCD algorithm specifically for the ReLU activation $\sigma(x) := \max\{x, 0\}$, which has been excluded in Theorem 4.1 due to Assumption 2. The difficulty in handling the ReLU activation is that it only takes non-negative values. For attaining zero loss for a hidden layer $\|\sigma(W_j V_{j-1}) - V_j\|^2$, we need to prevent $V_j$ from taking negative value due to this non-negativity. Therefore, we must exclude such situations by modifying Algorithm 1.

### 5.1 BCD FOR NEURAL NETWORKS WITH SKIP CONNECTION

As a solution to overcome the difficulty of ReLU activation, we consider ResNet (He et al., 2016) type networks, where the neural networks includes skip connection. With skip connection, the

objective function treated in BCD is given by

$$\min_{\mathbf{W},\mathbf{V}} F(\mathbf{W},\mathbf{V}) \coloneqq \sum_{i=1}^{n} \left[ (W_L V_{L-1,i} - y_i)^2 + \gamma \sum_{j=1}^{L-1} \|\sigma(W_j V_{j-1,i}) + V_{j-1,i} - V_{j,i}\|^2 \right],$$

where the loss of the hidden layer, $\gamma \sum_{j=1}^{L-1} \|\sigma(W_j V_{j-1,i}) + V_{j-1,i} - V_{j,i}\|^2$ differs from (4). We describe the modified algorithm in Algorithm 3. We use the notation $V_{j,i}^+ = \max\{V_{j,i}, 0\}$.

---

**Algorithm 3:** Block coordinate descent: ReLU

---

**Input:** $(W_1)_{ab} \overset{i.i.d.}{\sim} \mathcal{N}(0, 1/d_{in})$, $(W_j)_{ab} \overset{i.i.d.}{\sim} \mathcal{N}(0, 1/r)$ for all $j = 2, \dots, L$, $V_{0,i} = x_i$

1   $K$: max iteration, $K_{in}$: max inner iteration, $\eta_V, \eta_W^{(1)}, \eta_W^{(2)}$: step size;
2   $W_j \leftarrow \mathrm{SVB}(W_j)$ with inputs $s_1$, $s_2$, and $W_j$ for $j = 2, \dots, L-1$;
3   $V_{j,i} = \sigma(W_j V_{j-1,i}) + V_{j-1,i}$ for all $j = 1, \dots, L-1$ and $i = 1, \dots, n$;
4   **for** $k \leftarrow 1$ **to** $K$ **do**
5     **for** $i \leftarrow 1$ **to** $n$ **do**
6       $\big| \quad V_{L-1,i} \leftarrow \big( V_{L-1,i} - \eta_V \nabla_{V_{L-1,i}} \|W_L V_{L-1,i} - y_i\|^2 \big)^+$;
7     $W_{L-1} \leftarrow W_{L-1} - \gamma \eta_W^{(1)} \sum_{i=1}^{n} \nabla_{W_{L-1}} \|\sigma(W_{L-1} V_{L-2,i}) + V_{L-2,i} - V_{L-1,i}\|^2$;
8     **for** $j \leftarrow L-1$ **to** $2$ **do**
9       $W_j \leftarrow W_j - \gamma \eta_W^{(1)} \sum_{i=1}^{n} \nabla_{W_j} \|\sigma(W_j V_{j-1,i}) + V_{j-1,i} - V_{j,i}\|^2$;
10       **for** $i \leftarrow 1$ **to** $n$ **do**
11         **for** $k_{inner} \leftarrow 1$ **to** $K_V$ **do**
12           $\big| \quad V_{j-1,i} \leftarrow V_{j-1,i} - \gamma \eta_V \nabla_{V_{j-1,i}} \|\sigma(W_j V_{j-1,i}) + V_{j-1,i} - V_{j,i}\|^2$;
13         $V_{j-1,i} \leftarrow (V_{j-1,i})^+$;
14     **for** $k_{inner} \leftarrow 1$ **to** $K_W$ **do**
15       $\big| \quad W_1 \leftarrow W_1 - \gamma \eta_W^{(2)} \sum_{i=1}^{n} \nabla_{W_1} \|W_1 V_{0,i} - V_{1,i}\|^2$;

---

The initialization and update of $W_1, \dots, W_{L-1}$ are common in Algorithm 1 and Algorithm 3. However, there are several differences between the two algorithms in their update procedures. First, in Algorithm 3, we apply the non-negative projection $V \mapsto V^+$ for each $V_{j,i}$ after the inner loop finishes. This is required for the non-negativity of ReLU: to ensure the solvability of the equation $\|\sigma(W_j V_{j-1}) - V_j\|^2 = 0$. Next, we do not update $W_L$ in Algorithm 3. This is required to ensure the existence of $V_{L-1,i}$ satisfying $W_L V_{L-1,i} = y_i$ under the condition $V_{L-1,i} \geq \mathbf{0}$. To verify this, we first provide the following lemma.

**Lemma 5.1.** *Suppose that the vector $W_L$ has both positive and negative entries. Then, for any $y_i$, there exists a non-negative vector $V_{L-1,i}$ satisfying $W_L V_{L-1,i} = y_i$.*

This lemma implies that, to ensure the global convergence for arbitrary training label $y_i$, it is sufficient to check that $W_L$ has both positive and negative components. Clearly, such a situation will occur frequently as the with of the hidden layer $r$ increases. Indeed, by the symmetry of the Gaussian distribution, this probability is calculated as $1 - 2 \cdot \left(\frac{1}{2}\right)^r = 1 - 2^{-r+1}$. Additionally, we provide a high probability bound on the norm of the positive and negative components of $W_L$, which determines the convergence speed of the gradient descent.

**Lemma 5.2.** *Let $W_L^\top \sim \mathcal{N}(0, r^{-1} I_r)$, $w_+ \coloneqq \max\{W_L, \mathbf{0}^\top\}$, and $w_- \coloneqq \min\{W_L, \mathbf{0}^\top\}$. Then, for any $\delta > 0$, with probability at least $1 - 2\delta$, $\min\left\{\|w_+\|^2, \|w_-\|^2\right\} \geq \frac{1}{2} - \sqrt{\frac{8\log(2/\delta)}{r}}$ holds.*

Since it is not trivial that the similar inequality holds for each iteration when considering the update of $W_L$, we assume that $W_L$ is fixed during training for simplicity.

Similarly to the problem (4) considered in the previous section, we consider 1-dimensional outputs here. We then formally state the convergence result of Algorithm 3 applied to networks with ReLU activation and skip connections.

**Theorem 5.3** (Global convergence of BCD with ReLU activation). *We assume that there exists a constant $C_V > 0$ such that $\lambda_{\max}(V_j V_j^\top) \leq C_V$ for $j = 1, \dots, L-1$ during training. We denote $s \coloneqq \sigma_{\min}(X)$. Let $R_i = |W_L V_{L-1,i} - y_i|$ at the initial value of the objective function with*

*respect to the output layer, and define $R := \sum_{i=1}^{n} R_i^2$, $R_{\max} := \max_i R_i$, and $C_K := (4R_{\max}\eta_V + 5\sqrt{\epsilon})\left(\frac{3}{2}\right)^L$. Then, under $(s_1, s_2) = (0, \frac{1}{4})$, $\eta_V \leq \frac{1}{2\min\{\|w_+\|^2, \|w_-\|^2\}}$, $\eta_W^{(1)} \leq \frac{\eta_V^{-1}}{24\sqrt{r}C_V K}\left(\frac{2}{3}\right)^L$, $\eta_W^{(2)} \leq \frac{1}{2\cdot\max_i\|x_i\|}$, and*

$$K = \left\lceil \frac{1}{4\eta_V\min\{\|w_+\|^2, \|w_-\|^2\}}\log\left(\frac{3R}{\epsilon}\right)\right\rceil, K_V = \left\lceil \frac{3}{4\gamma\eta_V}\log\left(\frac{49(L-2)rnC_K^2}{3\epsilon}\right)\right\rceil, K_W = \left\lceil \frac{1}{4\gamma s\eta_W^{(2)}}\log\left(\frac{C_K^2}{\epsilon}\right)\right\rceil,$$

*it holds $F(\mathbf{W}, \mathbf{V}) \leq \epsilon$, where $\mathbf{W} = (W_1, \ldots, W_L)$ and $\mathbf{V} = (V_{1,1}, \ldots, V_{L-1,n})$ are the parameters obtained by the output of Algorithm 3.*

The proof can be seen in Appendix F. Thus, we obtain a global convergence guarantee of BCD for networks with ReLU activation.

## 6 NUMERICAL EXPERIMENT

In this section, we conduct numerical experiments to verify our theoretical findings. Particularly, we numerically confirm that BCD converges to a global minimum for monotonically increasing activation (Algorithm 1) and ReLU (Algorithm 3) using an artificial dataset.

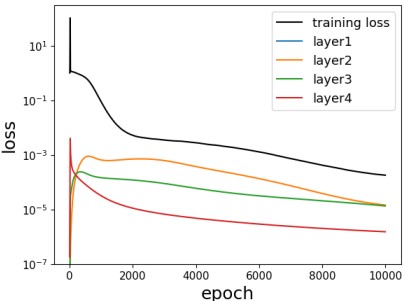 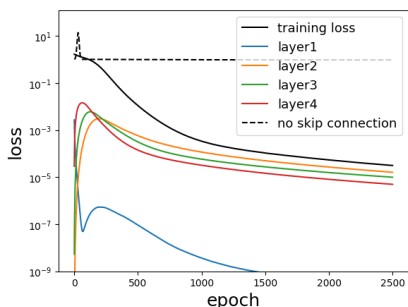

Figure 1: Loss of Algorithm 1 with LeakyReLU      Figure 2: Loss of Algorithm 3 with ReLU

### 6.1 MONOTONICALLY INCREASING ACTIVATION

First, we conduct a numerical experiment for a monotonicall y increasing activation. We apply Algorithm 1 to a neural network with four hidden layers, each with $r = 30$ nodes, and LeakyReLU activation $\sigma(x) = \max\{x, 0.5x\}$, which satisfies Assumption 2 with $\alpha = 0.5$ and $\ell = 1$. We prepare $n = 500$ training samples from a teacher network with a single hidden layer and the same activation. We set $d_{in} = 600$, sample $x_i$ from the normal distribution, and define $y_i$ as the output of the teacher network. For hyperparameters, we employ $K_V = K_V = 100$ and $\eta_V = \eta_W^{(1)} = \eta_W^{(2)} = 1$.

Figure 1 shows the result. The black line means the training error, i.e., $\frac{1}{n}\sum_{i=1}^{n}(f_{NN}(x_i) - y_i)^2$. Other lines represent the loss of $j$-th layer, i.e, $\sum_{i=1}^{n}\|\sigma(W_j V_{j-1,i}) - V_{j,i}\|^2$ for $j \in \{1, 2, 3, 4\}$. We can observe that the training error monotonically decreases while the losses for each layer remain small, which reflects our theoretical findings.

### 6.2 RELU ACTIVATION

Next, we experimentally examine BCD for ReLU activation using Algorithm 3. We apply Algorithm 3 to a neural network with for hidden layers, $r = 30$, ReLU activation and skip connection. Similarly to the monotinically increasing activation, we prepare a dataset with $n = 500$ and $d_{in} = 600$ using a teacher network. For hyperparameters, we employ $K_V = K_V = 100$ and $\eta_V = \eta_W^{(1)} = \eta_W^{(2)} = 1$.

Figure 2 shows the result. Like Figure 1, the black line means the training error. Other lines represent the loss of $j$-th layer, i.e, $\sum_{i=1}^{n}\|\sigma(W_j V_{j-1,i}) + V_{j-1,i} - V_{j,i}\|^2$ for $j \in \{1, 2, 3, 4\}$. We can observe the same convergence procedure here: the training error monotonically decreases and the losses for each layer remain small.

Additionally, we plot the training loss without using the skip connection as the dashed black line. While the training loss for BCD without skip connections does not decrease due to the difficulty of maintaining non-negativity, the skip connection drastically improves BCD training.

## 7 CONCLUSION

In this paper, we proposed a block coordinate descent (BCD) algorithm for training deep neural networks and ensured the convergence to global minima for networks with strictly monotonically increasing activation functions. We also derived a generalization bound using Rademacher complexity, ensuring both strong optimization and generalization performance. For ReLU activations, we introduced a modified BCD algorithm with skip connections and non-negative projection updates to ensure convergence. Empirical validation demonstrated the practical effectiveness of our algorithms for both monotonic and ReLU activations. Overall, this work advances the understanding of BCD in neural networks, offering provable convergence and generalization guarantees.

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
