# OpenReview forum: "Block Coordinate Descent for Neural Networks Provably Finds Global Minima"
_ICLR.cc/2025/Conference — Submitted to ICLR 2025_

### Official Review · Reviewer_Ph8B · 2024-10-31

**Soundness:** 2
**Presentation:** 3
**Contribution:** 1
**Rating:** 3
**Confidence:** 3

**Summary:**

This paper shows that under suitable assumptions and a specific design of the objective, block coordinate descent will always find parameters that have both arbitrarily small training loss and good generalization guarantee. The major assumptions this paper makes are that the dataset is full rank in the high-dimensional regime (where the input dimension >= number of data), and the activation is bijective. Optimality guarantee follows from the fact that the given objective is a sum of least squares, and optimizing least squares by gradient descent (or its variants) is possible. Generalization guarantee follows from the fact that they chose the hyperparameters appropriately to make the operator norm of the weights well-behaved, plus at the initialization they use singular value bounding. Finally, the paper extends the result to ReLU(which is not bijective) by adding skip connections, and show by numerical experiments that their claim holds.

**Strengths:**

1. The paper tackles an important problem and provides a new perspective that "block coordinate descent may be a better way to optimize".
2. The paper is easy to follow with sufficient insights and motivations so that the reader can understand the theorems and algorithms.

**Weaknesses:**

1. One critical issue that I did not understand is the following:

In this setting, do we "need" block coordinate descent? If so, why?

This question emerged because when the activation $\sigma$ is bijective (which is the case when Assumption 2 is satisfied), we could repeatedly solve a series of linear equations to find the optimal parameters $W_L, W_{L-1}, \cdots W_{1}$. Say $Y \in \mathbb{R}^{d_{out} \times n}, X \in \mathbb{R}^{d_{in} \times n}, W_L \in \mathbb{R}^{d_{out} \times r}, W_1 \in \mathbb{R}^{r \times d_{in}}, $and $W_i \in \mathbb{R}^{r \times r}$ for $i = 2, 3, \cdots L-1$. We could solve Y = W_LV to find the correct $V$, and then apply $\sigma^{-1}$, and then solve $\sigma^{-1}(V) = W_{L-1}V'$, and repeat this process until we find good parameters that fit $Y$ given $X$. Hence, without the complicated block coordinate descent algorithm, can't we train the network to global optimality?

2. Another weakness I find is the constant $C_V$. This paper assumes that the operator norm of $V_j$ is bounded by $C_V$, which is a convenient assumption to make many things simple (for example, it is used to prove that the weights $W_i$ behave well - which is used to prove generalization guarantees): Remark 4.2 says that $C_V$ can be expressed using $\eta_V$, $K$, and $K_V$, but I don't see a clear reason why. As $C_V$ is crucial in proving generalization bounds and well-behaving norms, clarifying this remark is I think needed.

3. The assumptions are somewhat restricted: $d_{in} \geq n$ seems restrictive, and $\sigma$ needing to be bijective is also quite restrictive. The network has a particular size (the weights are all square matrices in the intermediate layers). Also, it could be better if the authors could give an ablation study on singular value bounding: does this procedure help training, or is it a procedure that is of theoretical interest?

4. In numerical experiments, the trainning loss in Figure 1 does not seem to converge to 0 (it seems like it is decreasing, but it's hard to see that its a global optimum) - running this more to show that the algorithm actually finds a global minimum could be nice.

**Questions:**

1. In pg 21, there is a proof without any theorem. This could be a typo.

2. Addressing the weaknesses would be nice. If I see that block coordinate descent is beneficial, $C_V$ has a bound that only depends on the system parameters, I would definitely raise my score.

3. Assumption 2 should be made clearer. Subdifferential is only defined when $\sigma$ is convex. What if it is not convex? One suggestion is writing for any $u_1, u_2 \in \mathbb{R}$, $\sigma(u_2) - \sigma(u_1) \geq \alpha |u_2 - u_1|$. Another suggestion is assuming $\sigma$ is convex.

4. Is the intuition of adding skip connection for ReLU case trying to make ReLU into a bijective function?

---

> ### Author Response · Authors · 2024-11-22
>
> Thank you for your insightful feedback and comments. First, we would like to address the points you raised as weaknesses.
>
> - Do we "need" block coordinate descent? If so, why?
>
> A. As you mentioned, in our settings, where we employ the loss function (2) with strictly increasing activation, the optimal value of V can be obtained by directly solving the linear equation. However, there exist several limitations to employ such an approach. First, we consider the training without any device of loss function for simplicity, for example, including regularization term, which is more typical in the BCD literature. In such a case, we can no longer explicitly write down an analytical solution of $\mathbf{V}$. On the other hand, BCD has room for extension. Moreover, for the case with ReLU activation, where we employ the skip connection to ensure global convergence, we cannot solve $\mathbf{V}$ directly. These are the reasons why we are considering the BCD training.
>
> - How to obtain the bound on $C_V$.
>
> A. Thank you for pointing out an important issue. The bound on $C_V$ can be derived by the bound on $V_{j,i}$ during the training. We obtain the bound on $V_{j,i}$ as follows: when we provide the convergence guarantee with respect to the losses of hidden layers by updating $V_{j,i}$, we prove the linear convergence of them as we exhibited in Lemma C.7. This implies that the distance between $V_{j,i}$ and the optimal solution of inner loops can be bounded by that at the start of inner loops (maybe multiplied by a constant factor). Then, we only need to evaluate the difference in the optimal solution, which can be evaluated by using the system parameters.
>
> - The assumptions are restrictive.
>
> A. Thank you for pointing out the limitations of our results. As you mentioned, we impose several restrictions on the settings. First, assumption $d_{in}\ge n$ is imposed to ensure the existence of the solution of equation $\sigma(W_1X)=V_1$ where $W_1\in\mathbb{R}^{r \times d_{in}}$ and $V_1$ is a matrix whose $j$-th row is $V_{1,j}$, which is the loss with respect to the first hidden layer. This equation can be decomposed into that of each node, that is, $\sigma(w_kX)=(V_1)_k$, where $(V_1)_k\in\mathbb{R}^n$ is a $k$-th column of $V_1$. Thus, rank(X) = n is a sufficient condition to make this equation have a solution, so we assume this condition. The assumption on $\sigma$ is also restrictive, while it admits LeakyReLU activation. That is why we consider the extension to ReLU activation in section 5. Moreover, we discussed the extension to other activation in section B. Finally, the singular value bounding helps the convergence by improving the convergence rate of our analysis (intuitively, it improves the condition number of the loss). This improvement is not a theoretical one but also enhances the empirical convergence, as we observed in numerical experiments. We would like to add the ablation study of utilizing singular value bounding after completing the experiment. We appreciate your helpful suggestions for improving our results.
>
> - In numerical experiments, the training loss in Figure 1 does not seem to converge to 0 .
>
> A. Thank you for pointing out an issue with our experiment. We have conducted a numerical experiment with further epochs and updated the figure.
>
> Then, we would like to answer your questions.
>
> Q. In pg 21, there is a proof without any theorem. This could be a typo.
>
> A. The proof in pg 21 is that of Theorem 4.3. We have fixed this issue in the revised version. Thank you for pointing this out.
>
>
> Q. Assumption 2 should be made clearer.
>
> A. Thank you for pointing out the issues with our writing. We have written the sentence to
>
> If $\sigma$ is not differentiable, we assume that $\sigma(x_1)-\sigma(x_2)\ge \alpha(x_1-x_2)$ for any $x_1$, $x_2\in\mathbb{R}$.
>
> We have fixed this point. Please see the revised version.
>
> Q. Is the intuition of adding skip connection for ReLU case trying to make ReLU into a bijective function?
>
> A. There are several reasons to add the skip connection. One reason is that, as you mentioned, the skip connection makes ReLU bijective, ensuring global minima exists. Moreover, from the optimization perspective, the skip connection helps the training procedure avoid vanishing gradient problems, similar to the original motivation of the skip connection (He et al., 2016). Without a skip connection, the training in the loss with respect to hidden layers will stop if the pre-activation takes negative values. By employing the skip connection, we can ensure the linear convergence guarantee, as we show in the proof of Theorem 5.3.

---

> > ### Comment · Reviewer_Ph8B · 2024-11-23
> >
> > Thank you for the very detailed response and accomodating some of the things that I pointed out. However, I am still not convinced about the major weaknesses. Specifically,
> >
> > 1) My point in weakness 1 was that the problem that this paper tries to solve "is easy to train to global optimality" - by imposing bijectivity on $\sigma$. I understand BCD has room for extension, i.e. we could solve the regularized problem with BCD. However, this paper is not proving that BCD can train regularized neural networks to global optimality. Also, for ReLU with skip connections, we can construct a globally optimal solution by the following algorithm:
> >
> > We choose $V_{L-1}$ as a matrix that satisfies $W_LV_{L-1} = Y$, and choose $V_{L-2}$ as a matrix with full rank, $0 \geq V_{L-2} \geq V_{L-1}$ and choose $W$ to be a matrix that satisfies $\sigma(WV_{L-2}) = V_{L-1}-V_{L-2}$. As $V_{L-1} - V_{L-2}$ is positive, we can find such $W$ as $V_{L-2}$ is full rank.
> >
> > My concern is that the main claim that the authors propose, which is "BCD can provably find global minima in certain problems", sounds weak. That is because the "certain problems" that the authors are discussing has an inherent structure that makes finding global minima easy (because we can simply fit things perfectly). I would appreciate it very much if the authors could address this point.
> >
> > Also, while reading the ReLU part again, I couldn't understand line 405-406: why do we need to ensure the solvability of \lVert \sigma(W_{j}V_{j-1}) - V_{j} \rVert^2 = 0?
> >
> > 2. Thank you for clarifying. However, I am still not convinced. The reason is:
> >
> > $\eta^{(1)}_W$ depends on $C_V$. Hence, the $W$ s change due to the change in $C_V$ (they depend on $C_V$). I believe a crucial step in showing the stable behavior of $V$ is using the stable behavior of $W$, which is proven with the assumption that $C_V$ exists. Say $C_V$ does not exist. Can we prove that $W$ is stable?
> >
> > I think the logic here is that $W$ is stable over the training process, $V$ is stable for fixed $W$s in the training process, hence $V$ is overall stable and we can find $C_V$ with the system parameters. However, $W$ being stable is not derivable without the assumption that $C_V$ exists.
> >
> > I could have misunderstandings, so please correct me if I am wrong.
> >
> > 3) I tried to checked out the revised paper, but I couldn't find one. Where can I see the revised version?

---

> > > ### Author Response · Authors · 2024-11-25
> > >
> > > Thank you for raising your further questions. We would like to address them.
> > >
> > > Thank you for clarifying your concerns on weakness 1. Here, we would like to provide several comments on this.
> > >
> > > First, while we can obtain the optimal solutions by directly solving the linear equation, as you pointed out, it is not straightforward to ensure the global convergence of the gradient method. This is mainly caused by two reasons: (i) the existence of non-linearity results in a difficulty even if it is monotonically increasing. (ii) we need to evaluate how the optimality gap remaining in the gradient descent on the layer affects the loss with respect to the former layer. Please check the proof of Theorem 4.2 for more details on these points.
> > >
> > > Second, there is also room for extension to other networks. For example, regarding the multilayer perceptron, we admit networks with narrower widths in the latter layers, i.e., $d_{in}\ge d_1\ge\dots d_{L-1}\ge d_{out}$, where $d_j$ is the width of the $j$th layer. Moreover, it will be interesting future work that how our analysis can be extended to more modern architectures such as an attention layer (including how we can apply the BCD formulation). On the other hand, we cannot treat the situation $d_{j}<d_{j+1}$ for a specific $j$, which is inevitable due to the “solvability problem” similar to what we discuss in the ReLU part.
> > >
> > > Thank you for the clarification of your concerns.
> > >
> > > - **about the ReLU part**
> > >
> > > We intend to state that we need the solvability to ensure the **existence** of global minima. For the ReLU activation, if $V_j$ has a negative entry, no $W_j$ and $V_{j-1}$ can satisfy $\lVert \sigma(W_{j}V_{j-1}) - V_{j} \rVert^2 = 0$. This results in a situation where the parameters interpolating the labels do not exist, and we cannot ensure that BCD will achieve an arbitrarily small loss.
> > >
> > > 2. As you stated, our logic here is that the stable behavior of $W$ leads to the stable behavior of $V$. More precisely, we have shown that
> > > The singular values of W are within the interval $[\frac12,2]$ during the training.
> > > Based on the above condition on $W$, $V$ can exponentially converge to the global minima.
> > > Here, the point is that the analysis of the first condition depends on the $C_V$. To obtain this, we only need to ensure that $V$ will **not diverge** during the training. The exponential convergence nature of $V$ can ensure this. By employing a sufficiently small step size according to the norm of $W, V$ exponentially converges to global minima, and it will not diverge. We would appreciate you asking further questions about this point.
> > >
> > > 3. We have uploaded the revised paper and supplementary material. We believe that you can download them from the top of this page.

---

> > > > ### Comment · Reviewer_Ph8B · 2024-11-25
> > > >
> > > > I sincerely appreciate the authors for further clarification. I now can agree with the authors that even though this setup has an easy way to train to global optimality, maybe BCD methods could further have guarantees in training neural networks for different setup, leading extensions of this paper (e.g. regularized case, different architecture).
> > > >
> > > > Unfortunately, I am still not convinced about the clarification the authors made in point 2. As far as I understood, the logic is:
> > > > we know that $W$ is stable due to the fact that $C_V$ exists (the proof of Lemma C.4.), and use that fact to prove that $V$ does not diverge (due to the exponential convergence nature of $V$) and we deduce that $C_V$ exists-> which sounds like a circular logic to me.
> > > >
> > > > One straightforward thing the authors could do to clarify this point is giving the actual bound the authors said exists in Remark 4.2. (only using the system parameters).
> > > >
> > > > Also, I am curious why the loss does not decrease to 0 in Figure 1 (it looks like it is 1e-3 when the training is finished: though extra training could improve the loss value).

---

> > > > > ### Author Response · Authors · 2024-11-26
> > > > >
> > > > > Thank you for your further comments.
> > > > > Based on your comment on Remark 4.2, we have added the explicit bound on $C_V$ in Remark 4.2 and its derivation in Appendix D. We have uploaded the revised version, and we would appreciate it if you checked these points.

---

> > > > > > ### Comment · Reviewer_Ph8B · 2024-11-26
> > > > > >
> > > > > > Thank you so much for the revision. I am still having trouble understanding: so we first assume that $C_V$ exists, and use it to bound $||W||_{op}$(which is Lemma C.4) then bound $C_V$(which is you added in Appendix D).
> > > > > >
> > > > > > What I "don't" understand is, how do we know that $C_V$ exists in first place? We use the fact that $C_V$ exists in Lemma C.4. Is there a justification for this?

---

> ### Author Response · Authors · 2024-11-26
>
> Thank you for clarifying your concern.
> A key point concerning Lemma C.4 is that **we can choose the stepsize parameter $\eta_W^{(1)}$** according to C_V.
>
> Our logic here is that
> 1. If the singular values of $W$ are lower (1/2) and upper (2) bounded, we can ensure the stable update of $V$, which leads to global convergence.
> 2. According to the analysis of $V$, the required number of iterations $K$ and $K_V$ is determined. From these parameters, we can evaluate the bound on $C_V$.
> 3. By taking $\eta_W^{(1)}$ sufficiently small according to Theorem 4.2, we can ensure Lemma C.4.
>
> In other words, we need to choose $\eta_W^{(1)}$ sufficiently small to make the optimization of W stable. This is not circular logic because we can set $\eta_W^{(1)}$.
>
> Have this addressed your concern? We apologize for the confusion.

---

> > ### Comment · Reviewer_Ph8B · 2024-11-26
> >
> > I think I sort of understand what the authors are trying to do: precisely speaking, I suggest the following:
> >
> > First I would like to point out that we cannot choose sufficiently small $\eta_{(1)}$. The reason is: say the size of $V$ happened to be unbounded in the training process. No matter how small we choose $\eta_{(1)}>0$, there will be a very large $V$ that violates the assumption of $\eta_{(1)}$, and we cannot choose any positive stepsize this case. I am concerned about this case.
> >
> > What the authors probably can do is making $\eta_{(1)}$ depend on the timestep, i.e. make it adaptive with respect to the size of $V$. With this I think the math will be clear and things could be proved as the authors wanted.

---

> > > ### Comment · Reviewer_Ph8B · 2024-12-02
> > > **Final Decision**
> > >
> > > I understand that BCD could potentially be extended to general cases where simple fitting does not work, but on other hand still think simply fitting the network with BCD is not interesting enough. Also, I still think the assumption of $C_V$ is too strong, and I cannot find a way to write it in terms of only system parameters. The authors did explain the high level intuition (which now I sort of understand), it is not rigorous and I am skeptical if $C_V$ can really be bounded by only system parameters.
> > >
> > > I think my initial decision was fair, and decide to keep my score.

---

### Official Review · Reviewer_AAZM · 2024-10-31

**Soundness:** 1
**Presentation:** 2
**Contribution:** 2
**Rating:** 3
**Confidence:** 3

**Summary:**

The contribution is theoretical: establishing the global convergence for a specific block coordinate descent used to optimize neural networks. Indeed, The specific block coordinate descent is a particular layerwise optimization method. While it is not explicit, the global convergence result is established for the overparameterized regime, where the number of parameters $>$ the size of the training set.

**Strengths:**

- Studying the convergence of BCD is well motivated in the intro using the related literature
- Assumptions 1 and 2 are reasonable
- The theoretical settings is not far from the practical settings.
- If the analysis and statements were correct, global convergence of any algorithms on deep neural networks is a valuable result.

**Weaknesses:**

I think there are mistakes in the proof.

**Issue 1.** Check line 796 in the appendix. Recall notations $w'= w - \nabla $ where $\nabla = \eta X^\top D(\sigma(Xw)-Y)$. Line 796 proposes the following equation holds

$$\|| \sigma(Xw') - Y\||^2 = \|| \sigma(Xw) - X \nabla -Y\||^2 $$

The above equation only holds for a linear $\sigma$. However, the equation does not hold for a general $\sigma$ obeying Assumption 2. This discrepancy is significant, as the equation underpins the derivations used to establish the main theorems.

**Issue 2.**
In lines 802-803, it is claimed that $\||I- \eta \Xi X X^\top D \||_{op} \leq 1$
holds for a small positive $ \eta$. But why? If $A=\eta \Xi X X^\top D$ is not PSD, we can not ensure that the operator norm of  I - A  is less than one.

**Questions:**

- Does the specific BCD considered in this paper connect to layerwise training of deep neural networks? If so, this literature can be used to motivate the topic from a practical view.
- Is BCD cheaper than back propagation? Can you quantify the computational benefits of BCD?
- You mentioned your analysis in section C.2 (where I pointed out the mistake) is similar to [Yehudai and Ohad (2020); Frei et al. (2020)]. Can you be more specific about the similarity?

---

> ### Author Response · Authors · 2024-11-22
>
> Thank you for your insightful feedback. First, we would like to mention the issues you raise as mistakes.
>
> **Issue 1**
>  Line 796 causes a misunderstanding because of a typo. It should be
> $$
> 	 \|\sigma(Xw')-Y\|^2 = \|\sigma(Xw-2\eta XX^\top D(\sigma(Xw)-Y))-Y\|^2
> $$
> instead of
> $$
> 	 \|\sigma(Xw')-Y\|^2 = \|\sigma(Xw)-2\eta XX^\top D(\sigma(Xw)-Y))-Y\|^2
> $$
> in the original submission pdf (there was an extra “)” in the original version).
> We apologize for the confusion and have already fixed the point in the revised version.
>
> **Issue 2**
> As you pointed out, the matrix A is not PSD, so we need an additional argument as follows:
>
> We can express the matrix I-A as
> $$
> 	I-A = N(I-2\eta D^{\frac12}\Xi^{\frac12}M\Xi^{\frac12}D^{\frac12})N^{-1}
> $$
> with $M=XX^\top$ and $N=(\Xi^{-1}D)^{-\frac12}$. Since the matrix $2\eta D^{\frac12}\Xi^{\frac12}M\Xi^{\frac12}D^{\frac12}$ is PSD, this matrix will be contraction mapping for a sufficiently small $\eta$.
> Thank you for pointing out an important issue. We have already fixed the explanation in subsection C.2 by adding this argument with a detailed derivation.
>
> Moreover, we would like to answer the questions.
>
>
>
> Q. Does the specific BCD considered in this paper connect to layerwise training of deep neural networks?
>
> A. Our analysis does not directly cover the layer-wise training because we introduce the additional training parameters $\mathbf{V}$ and heavily rely on their update to ensure global convergence. However, there might be insights into the layer-wise connection by extending our analysis of the update of the weight matrix. That is promising future work, and we appreciate you raising that point.
>
>
>
> Q. Is BCD cheaper than back propagation?
>
> A. There is a memory/speed tradeoff between BCD and backpropagation. The decomposed objective function (2) enables us to implement the training procedures in distributed or parallelized manners at the expense of memory to record $\mathbf{V}$. Thus, BCD will provide faster training if we have enough memory.
>
>
>
>
> Q. Similar points to [Yehudai and Ohad (2020); Frei et al. (2020)].
>
> A. Our analysis in section C.2 can be seen as analyzing the dynamics of training networks with a single neuron. Such a setting has been theoretically explored in [Yehudai and Ohad (2020); Frei et al. (2020)], and they ensure similar convergence results while there are slight differences in their settings compared to ours. That is why we list these works as the related ones in section C.2.

---

> > ### Comment · Reviewer_AAZM · 2024-11-24
> > **Issue 2 is fundamental**
> >
> > I strongly believe that issue 2 is fundamental and breaks down all the proofs. For small stepsizes, we can only ensure the decay in loss only if gradient is not zero. Since the norm of gradient is shrinking to zero, the iterates will stop at a stationary point (local minimum or saddle point). I recommend authors to revisit the analysis carefully and quantitatively. How small stepsize has to be to ensure the decay in operator? Is it constant? If the gradient zero, then can still you get a norm less than one?

---

> > > ### Author Response · Authors · 2024-11-25
> > > **Supplement to the discussion about Issue 2**
> > >
> > > We thank reviewer AAZM for clarifying further concerns. We have revisited our proofs based on your original comment on issue 2.
> > > Here, we would like to provide an overview of why small stepsizes lead to exponential decay of the loss. First, the matrix I-A does **not** depend on the norm of the gradient since $A = 2\eta\Xi XX^\top D$ consists of
> > > - $\Xi$: a diagonal matrix defined by Proposition C.3, whose entries are lower (upper) bounded by the constant $\alpha$ ($\ell$),
> > > - $X$: $W_j$ or $V_j$,
> > > - $D$: a diagonal matrix with entry \sigma’(w^\top x), which is lower (upper) bounded by the constant $\alpha$ ($\ell$).
> > >
> > > Thus, by taking $\eta$ sufficiently small (its threshold is a **constant** only depending on \alpha, \ell, and X), we can ensure that the largest eigenvalue of the matrix $I-A$ can be bounded by a **constant** smaller than 1. The detailed derivation of the recursion formula is described in Section C.2. We would appreciate it if you asked further questions on this point.

---

> > > > ### Comment · Reviewer_AAZM · 2024-11-25
> > > > **A question about the correction**
> > > >
> > > > There is another issue in your derivation, please check derivation bellow and let me know if I am making a mistake here.
> > > >
> > > > $\|| I-A \||  =_{[line: 871]} \|| N (I - \eta  D^{1/2} \Xi^{1/2} M  D^{1/2} \Xi^{1/2} ) N^{-1}\||  \leq \|| N \|| \|| I - \eta  D^{1/2} \Xi^{1/2} M  D^{1/2} \Xi^{1/2} \|| \|| N^{-1}\|| \leq  \||N \|| \||N^{-1}\||$. Why does $ \||N \|| \||N^{-1}\||< 1$ hold for a small $\eta$?

---

> ### Author Response · Authors · 2024-11-25
>
> Thank you for raising a further possible issue with our derivation.
>
> We would like to note that
> $$
> 	\lVert N(I-2\eta D^{\frac12}\Xi^{\frac12}M\Xi^{\frac12}D^{\frac12})N^{-1}\rVert = \lVert I-2\eta D^{\frac12}\Xi^{\frac12}M\Xi^{\frac12}D^{\frac12}\rVert
> $$
> holds since the eigenvalues of the two matrices, $ND’N^{-1}$ and $D’$, are identical, where $D’=I-2\eta D^{\frac12}\Xi^{\frac12}M\Xi^{\frac12}D^{\frac12}$. This is because, for an eigenvector $u$ of $D’$, $Nu$ is an eigenvector of  $ND’N^{-1}$ with the same eigenvalue. Thus, we do not need to ensure $\lVert N\rVert \lVert N^{-1}\rVert<1$.

---

> > ### Comment · Reviewer_AAZM · 2024-11-25
> > **Follow-up question**
> >
> > You mean that $Nu$ is unit norm? How can the non-symmetric matrix $I-A$ have only *real* and *positive* eigenvalues?

---

> > > ### Author Response · Authors · 2024-11-25
> > >
> > > $Nu$ does not need to be the unit norm (in other words, we can consider $\tilde{u}=u/C$ with a constant C to let $Nu$ the unit norm). While the matrix $I-A$ is not symmetric, the matrix obtained by the transformation
> > > $$
> > > I-2\eta D^{\frac12}\Xi^{\frac12}M\Xi^{\frac12}D^{\frac12}
> > > $$
> > > is symmetric, and whose eigenvalues are real and positive by taking a sufficiently small $\eta$.
> > > Then, we can ensure that $I-A$ has the same eigenvalues as this PSD matrix since we obtain its eigenvector by $Nu$, as we discussed above.
> > >
> > > Thank you for the further question. Please do not hesitate to ask additional questions regarding this point.

---

> > > > ### Comment · Reviewer_AAZM · 2024-11-25
> > > > **More details on operator norm**
> > > >
> > > > Would you mind write down your definition of operator norm of a matrix and prove that the operator of $I-A$ is bounded by 1 using your argument about eigenvectors/eigenvalues? I can not see the connection for the **non-symmetric** matrix $I-A$.

---

### Official Review · Reviewer_pYFB · 2024-10-31

**Soundness:** 2
**Presentation:** 3
**Contribution:** 2
**Rating:** 5
**Confidence:** 4

**Summary:**

This paper studies the optimization of deep neural networks via a specific block coordinate descent (BCD) algorithm. Their algorithm introduces a set of auxiliary parameters $V_{l, i}$, defined to approximate the output of the $l$th layer on the $i$th data point, and considers a layerwise GD algorithm on both the $V_{l, i}$ and the original weights $W_l$. The main result, Theorem 4.1 is that their algorithm converges to a global min (i.e interpolating solution) when the activation functions are strictly monotonic; the authors also provide a generalization to the ReLU activation. Additionally, the paper proves a generalization bound for the output of the algorithm.

**Strengths:**

- It is an interesting question to prove that gradient based algorithms for neural networks can converge to global minima. The algorithm presented here is to the best of my knowledge novel, and this paper generalizes prior works which only proved stationary point convergence by proving global convergence.
- The paper is written quite clearly, and the proofs to the best of my knowledge appear sound.

**Weaknesses:**

- The paper provides an algorithm which, for a dataset with $n \le d_{in}$ data points, outputs a neural network which interpolates the data points. Crucially, however, this algorithm is not a standard optimization algorithm such as vanilla gradient descent. We already know that networks in the NTK regime converge to global minima, yet the NTK is insufficient as it fails to accurately capture generalization. Similarly, while the algorithm presented here also converges to a global minimum, there is no (theoretical or empirical) justification for why we expect this algorithm to generalize well, what its inductive bias is compared to vanilla gradient descent, or whether it can perform feature learning. More concretely, there is no notion of why this algorithm is "better" than NTK-based approaches. Such omissions greatly limit the significance of the paper.
- For example, experiments on a toy dataset which compare the test error between neural networks trained with vanilla GD, nets in the NTK regime, and training with BCD could support this claim. Another idea would be a specific sample complexity guarantee with an improvement over the NTK for learning some class of target functions with BCD.
- The paper does claim that BCD "provides favorable generalization performance" (line 20, abstract). The only justification for this is the generalization bound Theorem 4.3. However, it has been observed in the literature that such spectral generalization bounds are vacuous and fail to explain the success of deep neural networks [1, 2]. It is unclear why the bound in Theorem 4.3 should be interpreted as achieving "good" generalization.
- A more minor limitation is that the results only hold for $n \le d_{in}$, which is not satisfied in many standard deep learning settings.

[1] Fantastic Generalization Measures and Where to Find Them. Yiding Jiang, Behnam Neyshabur, Hossein Mobahi, Dilip Krishnan, Samy Bengio. https://arxiv.org/abs/1912.02178
[2] Uniform convergence may be unable to explain generalization in deep learning. Vaishnavh Nagarajan, J. Zico Kolter. https://arxiv.org/abs/1902.04742

**Questions:**

- More discussion on my above concerns re the significance of this work would be greatly appreciated.
- Theorem 4.1 assumes that $\lambda_{max}(V_jV_j^T)$ is bounded throughout training. This seems like a very strong assumption, and I don't see why one should expect this to hold (for example, it seems reasonable that this could scale with $d_{in}$ or $r$)). Can you comment on this point?
- A number of the algorithmic choices (use of different learning rates, different number of steps for different layers, etc.) seem a bit arbitrary, and so comments on these choices would also be helpful.

---

> ### Author Response · Authors · 2024-11-22
>
> Thank you for your insightful feedback. First, we would like to refer to the weakness you listed.
>
> - Comparison to the NTK regime.
>
> A. Thank you for pointing out an important issue. As you mentioned, the comparison of our convergence results to that of vanilla gradient descent derived by the NTK regime, for example. As we stated after Theorem 4.2, the BCD is potentially better than the NTK regime in the sense that the neural network trained by the BCD may have the feature learning ability. More precisely, in the analysis of the NTK regime, it is observed in both theoretical and experimental senses that the parameters of neural networks hardly move from their initial values. Our analysis does not fall into such a situation and allows for scenarios where the parameters undergo changes of $\Omega(1)$, which may lead to the feature learning ability. Moreover, while the NTK-type analysis requires the overparameterization of hidden layers, we can choose arbitrarily $r$.
>
> On the other hand, the more comparative analysis between BCD and the NTK regime by comparing their generalization error is a significant future work in investigating BCD. Thank you for suggesting a promising direction for our analysis.
>
> - The derived generalization error bound is not effective.
>
> A. As you mentioned, the uniform convergence type bound on the generalization error has been reported as not enough to explain the empirical success of deep learning. However, there are several reasons that deriving such a generalization error bound is worthwhile.
> First, as we stated in the paper, few results derive the generalization error bound for deep neural networks with arbitrary depth, ensuring their global convergence. This motivates us to believe that our analysis of BCD may be a promising standpoint from which to demystify the theoretical aspect of deep learning. Moreover, our results could be a cornerstone for constructing generalization error analysis of neural networks with optimization guarantee with further development of theory focusing on the generalization error bound on deep learning. Based on these reasons, our generalization error bound can be worthwhile in proceeding with the theoretical understanding of deep learning.
>
>
> - The assumption $n\le d_{in}$ is restrictive.
>
> A. Thank you for pointing out the limitations of our results. Assumption $d_{in}\ge n$ is imposed to ensure the existence of the solution of equation $\sigma(W_1X)=V_1$ where $W_1\in\mathbb{R}^{r \times d_{in}}$ and $V_1$ is a matrix whose $j$-th row is $V_{1,j}$, which is the loss with respect to the first hidden layer. This equation can be decomposed into that of each node, that is, $\sigma(w_kX)=(V_1)_k$, where $(V_1)_k\in\mathbb{R}^n$ is a $k$-th column of $V_1$. Thus, rank(X) = n is a sufficient condition to make this equation have a solution, so we assume this condition.
>
> Then, we would like to answer your questions.
>
> Q. The assumption $\lambda_{\max}(V_jV_j^\top)$ is restrictive.
>
> A. Thank you for pointing out an issue with the assumption. As you mentioned, the constant $C_V$ may depend on the parameters $r$ and $n$. The main purpose of this condition is to assume that the parameters $V_j$ will not **diverge** during the training, which can be ensured by using other parameters as we stated in Remark 4.2.
>
> Q. How to choose the hyperparameters.
>
> A. Thank you for the question. There are actually many choices for hyperparameters. We can choose different hyperparameters for each parameter as long as each learning rate and each number of epochs do not violate the conditions in Theorem 4.1 or Theorem 5.3. Just for simplicity, we assume the same learning rate and the same number of epochs for each element of \mathbb{W} and \mathbb{V} in algorithms 1 and 3.

---

### Official Review · Reviewer_qnaj · 2024-11-05

**Soundness:** 2
**Presentation:** 3
**Contribution:** 2
**Rating:** 5
**Confidence:** 3

**Summary:**

The paper proposes a block coordinate descent (BCD) training algorithm for deep networks and analyzes its convergence with respect to their weight parameters, which are proved to lead to a global minimum. This is done under monotonic increasing activation functions and ReLU (the latter with skip connections on the network. i.e., as ResNet). The paper also studies generalization only under monotonic increasing activation functions. A small number of simulations are provided to show the training convergence using the BCD algorithm.

**Strengths:**

- The paper is generally well-written. The presentation flows well.
- Remarks are added when appropriate, explaining connections with the literature.
- Algorithms are clearly written and the theoretical results are not difficult to parse.

**Weaknesses:**

-- Important issues:

- There is a circular argument in Theorem 4.1 and Theorem 5.3 which weakens the theoretical results. The assumption of the theorem states that $\\lambda_{min}(V_jV_j^\\top)$ is abounded across **all** the training by the constant $C_V$, but then this same constant $C_V$ is used to define the selection of the step-size $\\eta_W^{(1)}$ for the training. In other words, the authors are making an assumption on how the future behavior of the training will be, and then use this same assumption to establish the same training whose future behavior they are trying to study. This is a circular argument that makes not much sense. How can this be fixed?
I know that in Remark 4.1 it is stated that $\\lambda_{min}(V_jV_j^\\top)$ can be bounded differently by other parameters, which I think will be a better idea since that seems to get rid of the circular argument. However, no derivation is found of this other bound, and I believe it should be included. If such a bound is used, how would this change the theoretical results in the theorems and the proof derivations? There is no reason to believe that this different bound will work as well for the rest of the proof: new derivations are needed.

- The author establishes generalization results using classic Rademacher bounds found in Barlett et al. (2017). The problem is that the bound found by the author depends exponentially on the depth of the network! This has serious consequences: to achieve good generalization, according to the equation in Theorem 4.3, one would need to choose $n$, the sample size, larger than an exponential number of the depth $L$. This in turn means that the input neurons would also need to be exponentially large for deep neural networks (which has large $L$), since $d_{in}\\geq n$ for the optimization setting in the paper. In other words, the number of parameters and sample size required for a good generalization is exponentially large and impractical. This weakens the contribution of the paper, since such loose generalization bounds **already** exist on the literature. Given that such loose bounds exist in the literature for deep neural networks, why is it necessary to include it in the paper? I am not sure that the activation functions used in the paper are different from the ones existing in the generalization literature.

- Curiously, many of the works that are cited by the authors in lines 108-109 try to fix the exponential dependence on the depth. For example, it surprises me that the authors cite Golowich et al., 2018, which I believe is a work that introduces a new way to avoid such exponential dependence, and which can be used with Rademacher based analysis. Can the authors find a way to eliminate such exponential dependence drawing ideas from this work or related others?

- Another issue is that the paper does not propose any generalization bounds for the ReLU case with skip connections, i.e., for ResNets. Why is that? This makes the paper less complete. Can the authors derive such a generalization bound using similar techniques to the case of monotonic increasing activation functions? I don’t know whether or not generalization results exist for ResNets with ReLUs: if they don’t exist, the contribution by the authors would be strengthened.

- In line 235 it is claimed that the initialization done for the $V_{j,i}$ parameters leads to “faster convergence”. However, there is no theoretical result nor experimental result in the paper to support this claim. Why are the authors claiming this? This a strong claim.

- Related to the previous point, in line (64), it is mentioned that the “loss with respect to the output layer will decrease exponentially to zero”. However, how is this exponential convergence observed in the results of Theorem 4.1 and Theorem 5.3? Can the authors clarify this?

- The paper proposes a type of BCD algorithms (Algorithms 1 - 3). How does it compare to other types of BCD algorithms proposed in the literature, like the ones cited in the third paragraph of the Introduction? Why do the authors think these other works could not achieve global convergence? Could it be because of the particular algorithms used by these other works or could it be because of the proof techniques used by these other works?

- Line 049 claims that BCD has “more accessible landscapes to optimize”. What’s the basis for this claim? All the parts of the BCD loss are coupled with each other (look at equation (2) for example), so that doesn’t mean the energy landscape would be “more accessible to optimize”. It is still a non-convex optimization problem. The authors should explain what is meant by this, since I don’t see the right intuition with such statement.

- How does using activation functions that are monotonically increasing play a role in the derivations of the theoretical results? This would be important to know, even if a qualitative response is given. What happens when such condition is removed?

- There is a conflict of claims in the paper. In line 098, the authors claim that their analysis “does not require any overparameterization to ensure global convergence”. However, in line 273, the authors state that they have been working “in the overparameterized setting”. This is because they are in the regime $d_{in}\\geq n$, where the neurons in the input layer is equal or larger than the number of samples—which means that they are in the overparameterized regime. Can the authors clarify such conflict of statements and change the paper accordingly?
Do previous works on BCD also assume $d_{in}\\geq n$?

- For ReLU’s, why are skip connections important to assume for the proof techniques to work? What is the intuition behind it? Skip connections are arbitrarily introduced without any reference to how such new architectural change benefits the theoretical derivations compared to when there were no skip connections. There must be some motivation for it.

- In Section 6, the weights of the last layer are fixed after their initialization. How is it possible to achieve a global minimum, when the last layer weights are not being trained? Is there any intuition as to why this is the case? This would mean that every time we randomly initialize the neural network, there would exist a *new* global minimum whose weights on the last layer is the same as the one in the initialization—the existence of such result is not immediately intuitive and deserves an explanation.

- Regarding Figure 1 left, the layer 1 is seemingly missing from the plot. Also, the paper states that the last layer, i.e., layer 4, decreases to zero exponentially, but it is hard to see from the plot whether the layer 4 curve will go to zero: is it possible to show more epochs?

-- Other issues:
- Do people use BCD in practice? Contrary to works that study SGD, for example, or even perhaps GD, we know they are widely used in practice. What about BCD? The authors should add citations in Section 1 where BCD is used successfully in practical applications.
- Specify what kind of skip connections are implemented in the topology of the neural network in Section 5 (which is now a ResNet), since no description is explicitly found in the paper. Judging by the optimization equation, it seems that there is a connection from every neuron of the previous layer to the input of two layers ahead. Please, clarify. A mathematical equation description such as the one in line 136 would be helpful.
- The paper seems to suggest (see first paragraph of Introduction) that using the NTK is the only other existing method to formally provide optimization guarantees to neural networks. However, last year’s paper “Restricted Strong Convexity of Deep Learning Models with Smooth Activations” by Banerjee et al., published at ICLR 2023, proposes a new method for providing such guarantees without being on the NTK regime necessarily. This is a missing relevant citation.
- Typo in line 162: should be “field” instead of "filed”.
- What does it mean by a vector to have positive and negative “components” in Lemma 5.1? Do the authors mean “entries”?
- In Theorem 4.1 and Theorem 5.3, make explicit that $\\mathbf{W}$ and $\\mathbf{V}$ are the parameters obtained at the last iteration $k$ of their respective algorithms.

**Questions:**

Please, see all questions in the Weaknesses question.

---

> ### Author Response · Authors · 2024-11-22
>
> Thank you very much for taking the time to review our paper and providing us with valuable feedback. Here, we would like to comment on your concerns.
>
> - The assumption on the bounded $\lambda(V_jV_j^\top)$ will be a circular logic.
>
> A. Thank you for pointing out a possible issue with our results. First, we would like to mention that the choice of $\eta_W^{(1)}$ hardly affects the training dynamics of $V_j$. Why we require the bound on $\eta_W^{(1)}$ is that we need to provide lower/upper bounds on the minimum/maximum singular values on $W_j$ during the training. Indeed, as we have shown in Lemma C.2, the singular values of $W_j$ will stay in the region $[\frac12,2]$ during the training. Once this bound is obtained, we can ensure the convergence guarantee as Theorem 4.1 (and similarly as Theorem 5.3).
> Thus, our assumption on the bound on $\lambda(V_jV_j^\top)$ does not fall into a circular logic.
>
> - About the generalization error bounds exhibited in the paper.
>
> A. Thank you for enumerating several issues with our generalization error bounds:
>
> 1. Exponential dependency on the depth.
>
> 2. Whether we can obtain the refined bound using another existing literature.
>
> 3. We did not provide the bound for ReLU activation with a skip connection.
>
> For 1. and 2., we first would like to mention the paper (Golowich et al., 2018). It has provided the improved generalization error bound by exploring a generic technique. However, the product of the norms of each layer appears in the bound, which still results in exponential convergence on the network depth when applied to our settings. To the best of our knowledge, this dependence cannot be improved using the existing literature about the generalization error analysis of deep learning. Still, it is an important future work in the theoretical study of the BCD framework. On the other hand, we would like to state that almost all existing works on the generalization error analysis of multi-layer neural networks are conducted without an optimization guarantee. Our analysis is novel because we give the generalization error bound to trained neural networks.
>
> For 3., several existing works, such as (He et al., 2020), provide the generalization error bound on ResNet. While we can derive the bound by using these results combined with Theorem 5.3, we have omitted it just for simplicity. Thank you for your suggestion to improve the significance of our paper.
>
> He, Fengxiang, Tongliang Liu, and Dacheng Tao. "Why resnet works? residuals generalize." IEEE transactions on neural networks and learning systems 2020.
>
> - About the initialization of $V_{j,i}$ and the exponential convergence of the loss.
>
> A. First, the exponential convergence of the loss with respect to the output layer can be seen in the choice of $K$. To obtain the $\epsilon$ loss, we only need to loop outer loops for $K=\frac{2}{\eta_V}\mathcal{O}(\frac{3R}{\epsilon})$. When we focus on the dependency on $\epsilon$, this states that the loss exponentially decreases to zero. Please see the proof for a detailed derivation of this result.
>
> Our initialization scheme of $V_{j,i}$ helps us take small $K_V$, the number of inner iterations. In Lemma C.7, we have shown that the loss will exhibit exponential convergence with the exponent $-\gamma\alpha^2\ell\eta_V$ when we update $V_{j,i}$. Based on this lemma, we decide the enough number of inner iterations $K_V$ in Theorem 4.2 (similarly in Theorem 5.3).
>
> - Why do the authors think these other works could not achieve global convergence?
>
> A. We would like to note that we do not intend to argue that other BCD works could not achieve global convergence. While their algorithms may be able to converge to the global minima, they only ensure the convergence to stationary points, not to the global minima. Our motivation is to show that BCD **provably** converge to the global minima, and to state such a result, we employ a relatively simple algorithm compared to existing works in this paper.
>
> - The basis of the claim “BCD has more accessible landscapes to optimize”.
>
> A. As we mentioned in the introduction, the loss landscape of deep learning is highly non-convex, and hence, its convergence guarantees are limited to such as an NTK-type analysis. Compared to it, each term in (2) has a preferable landscape because of its simple structure, which can be seen as the sum of the loss with respect to each neuron. Indeed, in Lemma C.7, we can ensure exponential convergence with respect to each term during the training. This simple structure obtains this: since the loss function consists of linear transformation and monotonically increasing non-linear activation, we can provide a convergence guarantee with a slight modification to linear regression.

---

> > ### Author Response · Authors · 2024-11-22
> >
> > - The conflict of the statement on overparameterization.
> >
> > A. We apologize for the confusion. The statement “does not require any overparameterization to ensure global convergence” intends that we do not require the overparameterization on hidden layer, which is usually assumed in the optimization theory of deep learning such as the NTK regime. We already have fixed this point in the revised version.
> >
> > - The motivation of introducing skip connections.
> >
> > A. There are several reasons to add the skip connection. One reason is that the skip connection makes ReLU bijective, ensuring global minima exists. Moreover, from the optimization perspective, the skip connection helps the training procedure avoid vanishing gradient problems, similar to the original motivation of the skip connection (He et al., 2016). Without a skip connection, the training in the loss with respect to hidden layers will stop if the pre-activation takes negative values. By employing the skip connection, we can ensure the linear convergence guarantee, as we show in the proof of Theorem 5.3
> >
> > - How is it possible to achieve a global minimum, when the last layer weights are not being trained?
> >
> > A. Thank you for the important question. We can say that the fixed layer weights determine the optimal “pre-output feature” $V_{L-1,i}$, making the output perfectly interpolating labels. According to this optimal feature, we train the network parameters other than the output layers to reconstruct this. Thus, we can say we train the pre-output feature **instead of** the output layer weights. This is why we don’t need to train them.
> >
> > - About Figure 1.
> >
> > A. Thank you for pointing out issues with our experiments. We have conducted an experiment with further epochs, and we have already replaced the figure.
> >
> > Moreover, we appreciate your findings of several typos and suggestions related to work and rewriting. We have already fixed the points. Please check the revised version. We appreciate any further concerns.

---

### Meta-Review · Area_Chair_JoGk · 2024-12-18

**Metareview:**

This paper studies a Block Coordinate Descent (BCD) method for training deep neural networks with strictly monotone activations and high dimensional input. By studying the proposed BCD method on a “lifted” problem involving additional auxiliary parameters, the authors show convergence of the method to global minima. The authors also show a generalization error bound based on Rademacher complexity. The authors also consider ReLU networks and prove global convergence of a modified BCD algorithm.

The paper is well-written, but a couple of technical issues were raised by the reviewers and were not resolved during the discussion phase. One is a possible circular argument in assuming $\lambda_{\max} (V_jV_j^T) \leq C_V$ for some constant $C_V$, and the other is a possible mistake in bounding the operator norm of a non-symmetric matrix to claim contraction. The authors responded to these issues but the reviewers found the clarification insufficient. With these technical issues remaining, it is deemed that the submission could use a thorough revision before being considered for publication. Therefore, I recommend rejection at this time.

**Additional Comments On Reviewer Discussion:**

Other than the ones included in the summary above, issues raised by the reviewers raised by the reviewers include
- the possible vacuity of the Rademacher complexity upper bounds,
- the restriction that $d_{in} \geq n$ (which seems unavoidable due to the proof techniques).

Despite the authors' responses, I believe these downsides still stand, although they were not the critical reasons for my recommendation.

---

### Decision · Program_Chairs · 2025-01-22

Reject